# The Role of Electrochemotherapy in Managing Cutaneous Metastases from Breast Carcinoma: A Narrative Review

**DOI:** 10.3390/biomedicines13102345

**Published:** 2025-09-25

**Authors:** Francesco Russano, Davide Brugnolo, Paolo Del Fiore, Marco Rastrelli, Luigi Dall’Olmo, Simone Mocellin

**Affiliations:** 1Soft-Tissue, Peritoneum and Melanoma Surgical Oncology Unit, Veneto Institute of Oncology (IOV), 35128 Padua, Italy; francesco.russano@iov.veneto.it (F.R.); simone.mocellin@unipd.it (S.M.); 2Department of Surgery, Oncology and Gastroenterology (DISCOG), University of Padua, 35128 Padua, Italy

**Keywords:** breast cancer, BC, electrochemotherapy, ECT, cutaneous metastasis

## Abstract

Cutaneous metastases from breast carcinoma represent a debilitating complication of advanced disease progression, significantly impacting patients’ quality of life. Electrochemotherapy (ECT), which combines cytotoxic drugs such as bleomycin or cisplatin with electric pulses to enhance cellular permeability, has shown promising efficacy as a local treatment for these lesions. **Objectives:** This literature review examines the principles of ECT, its mechanisms of action and the clinical outcomes in patients with primary breast cancer. Across clinical series, patient-level ORR typically ranges from ~70–90% and CR up to ~58% at 6–12 weeks, with lower rates in larger (>3 cm) or deeper lesions. ECT is safe, well tolerated, and potentially synergistic with other systemic therapies. However, its efficacy is influenced by factors such as lesion size, tumor receptor status, and prior radiotherapy. Optimizing patient selection, standardizing treatment protocols, and developing combination approaches with immunotherapy or targeted therapies represent key future directions to improve clinical outcomes.

## 1. Introduction

Breast cancer (BC) is one of the most diagnosed and widespread cancers globally, and with an estimated 2.26 million cases recorded in 2020, it remains a significant cause of mortality among women [1]. BC is a heterogeneous disease influenced by both genetic and environmental factors, and it exhibits considerable molecular and histological diversity that necessitates varied therapeutic approaches [2,3]. Prevention and early detection campaigns in recent years have enhanced the curability of BC, though mastectomy remains the gold standard treatment. Despite advances in treatment, breast cancer retains a high potential for metastasis, commonly spreading to distant organs such as bone and the lungs, brain, and liver [4]. It is also the tumor most frequently responsible for skin and subcutaneous metastases within 10 years post-mastectomy [5,6]. While cutaneous metastases do not significantly impact overall survival, they severely affect the quality of life, causing pain, ulcers, infection, and psychological distress. Systemic treatments often fail due to side effects and the development of resistance to concurrent systemic therapies [7]. In recent years, electrochemotherapy (ECT) has emerged as an effective treatment for skin metastases derived from breast cancer and other solid tumors [8,9,10,11]. This technique combines the administration of cytotoxic, non-permeable chemotherapeutic agents, such as bleomycin and cisplatin, with the electroporation of tumor cells, thereby enhancing drug delivery [12]. Cutaneous metastases from breast cancer represent a rare clinical manifestation of the disease that indicate an aggressive disease progression. The overall incidence ranges from 0.6% to 10.4% among cancer patients, accounting for approximately 2% of all skin cancers. Approximately 10% of breast cancer patients develop distant metastases [4] and it is generally considered an incurable disease [13,14]. Cutaneous metastases occur in about 23.9% of patients with advanced breast cancer and they mostly involve chest and abdomen, though they can also appear on limbs and on the head and neck regions [15,16]. These metastases typically spread through lymphatic, hematogenous, or contiguous routes [17]. Cutaneous metastases are generally a secondary manifestation of advanced breast cancer but may also be the first sign of the disease. Most of them arise from adenocarcinomas and are commonly localized to the anterior chest [18]. It has been noted that patients with only cutaneous metastases generally have a better prognosis compared to those with stage IV breast cancer involving visceral metastases [19].

Cutaneous metastases from breast cancer represent a significant clinical and healthcare challenge, given their relatively high incidence among patients with advanced disease. Although their presence does not substantially affect overall survival, they are associated with considerable morbidity and have a profound negative impact on patients’ quality of life. Among the available local and systemic treatment options, electrochemotherapy (ECT) has emerged as one of the most effective therapeutic strategies for cutaneous breast cancer metastases. Clinical evidence consistently demonstrates high objective and complete response rates, particularly in patients with smaller and superficial lesions, alongside a favorable safety profile. Complication rates are generally low, with adverse events limited to manageable local reactions, further supporting the use of ECT as both an effective and well-tolerated treatment.

## 2. Methods

### 2.1. Literature Search Strategy

We conducted a literature search on PubMed/MEDLINE, Embase, Scopus, the Cochrane Library, and ClinicalTrials.gov from January 2001 to January 2025, with no language limits using keywords: “*breast cancer*”, “*cutaneous*/*skin*/*subcutaneous*/*chest wall involvement*”, and “*electrochemotherapy*/*electroporation*”. Records were deduplicated across sources prior to screening. To minimize publication bias, reference lists of the included articles and relevant reviews were manually screened for additional eligible studies.

### 2.2. Eligibility Criteria

We included original clinical studies enrolling adults with histologically confirmed breast carcinoma and cutaneous and/or subcutaneous metastases treated with electrochemotherapy (ECT) using bleomycin or cisplatin administered intravenously or intratumorally according to recognized operating procedures. A comparator arm was not required. Studies had to report at least one clinical endpoint (e.g., complete response, objective response, local control/durability, safety, or patient-reported outcomes). We considered randomized trials and prospective or retrospective cohorts (single- or multicenter). We excluded reviews, editorials, letters, conference abstracts; case reports or series with <5 patients; mixed-histology cohorts without breast-specific outcomes; and studies lacking clinical endpoints or where ECT was not a principal component. Only primary clinical studies contributed to Table 1 and to the qualitative synthesis.

### 2.3. Study Selection

Two independent reviewers screened all titles and abstracts. Full-text articles of potentially relevant studies were retrieved and assessed for eligibility. Any discrepancies were resolved through discussion or consultation with a third reviewer.

### 2.4. Data Extraction and Management

From each included study, the following data were extracted: authors, year of publication, study design, sample size, patient characteristics, type and size of cutaneous metastases, treatment protocols, outcome measures, and reported adverse events. Extraction was performed independently by two reviewers and cross-checked for accuracy.

### 2.5. Synthesis Approach

Given clinical and methodological heterogeneity—particularly in lesion burden/size, prior radiotherapy, response definitions, and denominators—we performed a qualitative synthesis without meta-analysis.

## 3. Electrochemotherapy: Principles and Mechanisms of Action

Electrochemotherapy (ECT) emerged in the late 1990s in both the United States and Europe as a highly effective, minimally invasive locoregional treatment for cutaneous and subcutaneous tumors. This therapeutic approach combines the administration of chemotherapeutic agents, most commonly bleomycin or cisplatin, with the application of locally delivered electric pulses to enhance drug uptake by tumor cells [34]. The electric pulses transiently permeabilize the plasma membranes of tumor cells through a process known as electroporation, thereby facilitating the intracellular delivery of otherwise poorly permeant cytotoxic drugs. As a result, the cytotoxic efficacy of these agents is significantly increased. Chemotherapeutic agents can be administered either intravenously (IV) or intratumorally (IT), depending on the tumor characteristics and treatment goals. Bleomycin is suitable for both systemic and local administration, whereas cisplatin is typically administered intratumorally to achieve optimal therapeutic concentrations at the tumor site. When bleomycin is administered systemically, electric pulses must be synchronized with the drug’s pharmacokinetic peak, which typically occurs 8–28 min post-injection in humans. In contrast, for intratumoral administration, electric pulses are usually applied within 1–10 min of drug injection to maximize therapeutic efficacy. The therapeutic activity of ECT involves multiple synergistic effects. Bleomycin induces double-strand DNA breaks, leading to mitotic catastrophe and subsequent cell death in dividing cells, while quiescent cells are less affected. Additionally, ECT induces transient local ischemia and permanent vascular disruption, contributing to decreased tumor perfusion and promoting drug retention within the tumor microenvironment. Notably, ECT appears to stimulate an immune-mediated response against tumor cells. The electroporation-induced release of tumor-associated antigens into the extracellular space may promote systemic antitumor immunity. This immune activation can potentially be amplified by adjunctive treatments with immunomodulatory agents, offering promising perspectives for combination therapies. The abscopal effect represents an area of growing interest. Although traditionally considered rare and unpredictable, the abscopal effect refers to the systemic antitumor response observed at sites distant from the primary treatment area. In ECT, this phenomenon may be triggered by the immunogenic cell death of tumor cells, leading to the release of tumor antigens and danger-associated molecular patterns (DAMPs). These molecules facilitate antigen presentation and the activation of cytotoxic T lymphocytes, which can subsequently recognize and destroy metastatic or untreated tumor cells elsewhere in the body. Emerging evidence suggests that combining ECT with immune checkpoint inhibitors or other immunotherapies may enhance the likelihood and magnitude of such systemic immune responses, thus broadening the therapeutic potential of ECT beyond local tumor control.

Although ECT can induce immunogenic cell death and theoretically trigger abscopal effects, the clinical evidence in breast cancer does not yet demonstrate durable systemic responses. Most published studies remain small, heterogeneous, and largely focused on local tumor control. This underscores the gap between mechanistic expectations and observed clinical outcomes, highlighting the need for prospective trials to evaluate the systemic immunological impact of ECT.

Standard procedures for the use of ECT including protocols for the administration of chemotherapeutic agents follow the guidelines of the European Standard Operating Procedure of Electrochemotherapy (ESOPE 2006), with updates over the years that have enabled the introduction and dissemination of the technique [35]. In recent decades, ECT has gained acceptance in oncology due to its minimal invasiveness and effectiveness in selectively targeting tumor cells [7]. It is also used to treat patients with unresectable tumors to ease pain or advanced stage tumors where surgical intervention is not feasible. In addition, ECT is employed to treat sensitive areas such as nerves and blood vessels to minimize the risk of adverse effects [36].

ECT has shown promising results in oncology, particularly in the treatment of cutaneous and subcutaneous tumors [37,38,39]. Skin metastases from basal cell carcinoma (BCC), squamous cell carcinoma, melanoma, Merkel cell carcinoma, head and neck cancer and angiosarcoma have been treated [40,41,42,43,44]. In addition, good results have also been reported in the treatment of deep visceral tumors and in the endoscopic treatment of colorectal cancer [45], and it is noteworthy that in recent years, the utilization of ECT for the treatment of skin metastases from BC has made significant progress [29]. The primary mechanism of action of electrochemotherapy (ECT) is summarized in Figure 1.

### Efficacy of Electrochemotherapy in Cutaneous Metastases of Breast Carcinoma

BC is a genetically and molecularly heterogeneous cancer with variable aggressiveness depending on tumor subtype, disease stage, and environmental factors [2]. Due to the high incidence, BC is the subject of extensive research, particularly regarding novel local and systemic treatment strategies.

Cutaneous and subcutaneous metastases represent a frequent clinical challenge, occurring in 5–30% of patients depending on disease stage. These lesions can present as single or multiple nodules, located most commonly on the breast, trunk, arms, legs, or back. Symptoms range from asymptomatic presentations to painful, bleeding, or infected lesions, often significantly impairing quality of life [32].

For limited skin and subcutaneous metastases, surgical excision remains a valid therapeutic option. However, when lesions extend or ulcerate around the mastectomy site, they pose a considerable challenge for surgical management and are associated with a marked decline in quality of life.

Targeted in situ treatment of skin lesions with electrochemotherapy (ECT) has proven to be highly effective, with significantly reduced side effects in comparison with other therapeutic options. It has been shown to be effective when used in conjunction with systemic cytotoxic therapies in high-risk patients, without increasing the toxicity [28].

In selected cases of recurrent BC leading to cutaneous lesions, where the patient refuses surgical intervention, ECT has proven to be an effective alternative [47].

The occurrence of skin metastases from BC varies with rates ranging from 5% in the general population to 30% in the later stages of the disease. Several groups have conducted retrospective studies to evaluate the overall success rate. Notably, a 2014 retrospective study involving 55 patients with skin metastases from BC and treated with ECT reported an overall complete response (local tumor control) rate of 40%. In addition, the study revealed that older patients (≥70 years) were significantly more responsive to the treatment compared to those under 70 years old [24], suggesting that ECT may represent an optimal therapeutic option in elderly patients often excluded from systemic therapies. There were no significant differences in the complete response rate according to the biological characteristics of tumor.

Another study involving ECT in the treatment of skin metastasis of various origins reported complete effectiveness in about half of the patients, with the best overall response for sarcoma (75%) and 46% for breast cancer, which is consistent with response rates previously documented in malignant melanoma cohorts. However, it demonstrates that the effectiveness depends on the tumor entity due to the fact that retrospective studies have limitations that do not allow for an accurate selection of both patients and tumor type (origin, stage, and previous treatments), providing only an indication of the treatment’s effectiveness with ECT, whose success rate could prove to be much higher [26]. According to a comprehensive literature review from 2017 that examined studies on the effectiveness of ECT for cutaneous and subcutaneous metastases from BC between 2011 and 2016, the overall response rate reached 76% [7,26].

A detailed systematic literature review was recently conducted by Ferioli and colleagues, analyzing the available evidence on the use of ECT for cutaneous metastases from BC. Retrospective and prospective studies were considered, but with strict inclusion criteria, such as clear parameters for patient selection and treatment methods, while excluding articles that did not report toxicity values or reviews, meta-analyses, case reports, etc., to make the results as comparable as possible. Essentially, the result is that ECT is well-tolerated and effective in terms of response to cutaneous metastases from BC, especially in lesions at a less advanced stage or smaller than 3 cm in diameter, where other local or systemic therapies had failed, with a CR observed in 46.2% of patients. Larger lesions, on the other hand, seem to be associated with a poorer tumor response. Toxicity, however, is still well manageable [48].

In the INSPECT Experience, the efficacy of ECT in breast cancer (BC) was analyzed in 171 patients treated between 2010 and 2020, stratified by receptor status. The study showed excellent responses in all subgroups examined, independent of conditions, age, time, and diagnosis. The best responses were obtained for smaller lesions. Response was assessed at 2 months. The objective response (OR) was 86% in HER2+ patients, 80% in HR+, and 76% in TN (*p* = 0.8664). Lesion size > 3 cm was the only negative prognostic factor. ECT was well tolerated, with grade 1 or 2 reactions and similar trends across the three groups. Local progression was observed in 16%. A significantly lower local progression-free survival (LPFS) was noted for TN tumors (78% vs. 81% vs. 61%), but with lower overall survival (OS) for TN tumors [32]. It is interesting to note that within the cohort included in the study, the geriatric population (over 90 years old) achieved results similar to the general population in terms of efficacy and safety, thus making ECT a treatment applicable even to the elderly population [49].

Similar results were also observed in the pan-European registry by the research group of Clover et al., who analyzed 987 patients and reported an overall response rate of 85%, with a 70% complete response at lesion level. When analyzing the subgroup of patients with cutaneous metastases from breast carcinoma, they observed a CR rate ranging from 77% to 62% [30], confirming high reproducibility of ECT results.

In the study by Grischke and colleagues [28], the concomitant use of systemic therapy (including estrogen therapy, chemotherapy, or monoclonal antibodies) with ECT according to ESOPE guidelines was analyzed in 33 patients with metastatic cutaneous (MC) breast cancer. The study achieved an objective response in 90% of cases without increasing local or distant toxicity, demonstrating the safety of integrating ECT into systemic treatment regimens.

Morley et al. [39] in their review analyzed 29 studies with a total of 1503 patients with cutaneous metastases of various origins. They found a complete response rate of 46.6% and an objective response rate of 82.2% (a reduction of at least 30% in the sum of lesion diameters) with statistically better outcomes for small tumors evaluated according to the RECIST method (greater than twice the probability of achieving a complete response). Adverse events were also limited, with one reported case of severe skin ulceration.

In the review by Wichtowski et al. [7], data similar to those from previous studies were reported, and they also found that positivity to estrogen receptors better correlates to ECT response. In the study by Cabula et al. [25], 113 patients were included, with an objective response rate (OR) of 90.2% and a complete response (CR) rate of 58.4%. Disease-free survival (DFS) was 86.2% at 12 months and 96.4% in patients with a CR. Factors significantly influencing the response rate included lesions <3 cm, absence of visceral metastases, low Ki67, and ER+. Reported complications included acute pain, grade 3 ulcers in 8% of cases, and grade 2 hyperpigmentation in 8.8%. Benevento et al. [21], analyzed 12 patients with a median follow-up of 210 days, achieving a CR in 75.3% and an OR of 92.3%. No severe complications were reported (one patient reported pain at the treatment site, and another reported superficial ulceration). In contrast, in the study by Matthiensen et al. [22], a CR was observed in only 8% of patients, with a partial response (PR) of 8% and stable disease in 76%. However, it is important to note that the patients in this study had cutaneous metastases >3 cm, confirming that lesion size significantly affects the response rate.

In the study by Campana [12], 125 patients with cutaneous metastases from melanoma, breast cancer, head and neck cancers, non-melanoma, and Kaposi’s sarcoma were analyzed. A CR of 58.4% was observed, with a significantly higher response rate in patients with metastases smaller than 3 cm, consistent with previous studies. Adverse events were mild, with pain at the treatment site in 10% of patients and mild dermatological reactions in 8%. They also found that there was a transient decrease in the dimensions of pain/discomfort and mobility and a persistent decline in self-care and usual activities [23]. Campana [50] also conducted another prospective study on 376 patients with MC from various origins, observing a CR rate of 50% and an OR of 88%. As in the previous studies, smaller metastasis size was predictive of complete response. In another study conducted by Matthiesen et al. [51], 90 patients with progressive cutaneous metastases undergoing systemic treatment after extended mastectomy were subjected to bleomycin-based ECT. A CR rate of 50% and a PR rate of 21% were observed. Stable disease in 16 (18%), and progressive disease in 7 (8%) [51].

The experience at our center was evaluated in the article by Russano et al. [31], which examined a large series of patients with metastatic cutaneous breast cancer treated with ECT between 1982 and 2017. This study examined the characteristics that may influence patient response. It was observed that a BMI lower than 22.5 kg/m^2^ and a body surface area lower than 1.77 m^2^ were associated with a higher response rate. Additionally, advanced stages were associated with lower efficacy, likely due to the more aggressive nature of the tumor. Reduced efficacy was observed in ER+ patients, while greater efficacy was seen in PgR− patients, HER2− patients, and those with higher-grade tumors. In this study, lesion size was also found to be predictive of a complete response to treatment. Previous treatment with radiotherapy (RT) was associated with a lower response to ECT, while post-ECT RT treatment may increase efficacy.

When comparing response rates across different histological subtypes of breast cancer with cutaneous metastases, it is observed that Luminal A and B tumors exhibit higher responsiveness. This finding was confirmed by a prospective multicenter study [33] involving 195 patients, which reported a complete response rate of 55%, a partial response rate of 27%, and an overall objective response rate (ORR) of 82%. When the data were stratified by molecular subtype, a significantly higher ORR was observed in patients with Luminal A and Luminal B tumors compared to those with triple-negative breast cancer, with ORRs of 88%, 90%, and 66%, respectively.

Less favorable results were reported in the study by Kreuter et al. [26], which included 56 patients with MC from various cancers (melanoma, breast cancer, squamous cell carcinoma, cutaneous lymphoma, or sarcoma). This study found an overall response rate of 44.6% and a complete response rate of 10.7%. The progression rate was 42.9%. Symptom improvement was observed in 25% of patients, with 27% reporting a reduction in lesion exudate, 25% a reduction in odor, and 18% a reduction in bleeding. Systemic effects were reported by 48% of patients, who complained of nausea and dizziness. Additionally, 3.6% reported pain at the treatment site, 5.3% superficial necrosis, 16% deep necrosis, and 8.9% bleeding and local infections. No severe adverse effects were reported. The lower response rates compared to other studies are likely due to the large size of the treated cutaneous metastases.

Intermediate results were instead observed by Bourke et al. in a cohort of 24 patients, reporting an objective response in 79.7%, with a complete response in 64.3%. A total of 19.3% of patients did not respond, while 1% showed disease progression [27]. These data are nevertheless affected by the limited number of cases included in the analysis.

ECT is also described in the literature for palliative purposes for non-resectable metastases. A notable case report [52] involved a 72-year-old woman with advanced breast cancer, cutaneous ulceration, and pectoral muscle infiltration. After rejecting therapy involving a skin flap, the patient was treated with ECT using bleomycin (15 mg/m^2^) and 101 pulses at 5000 Hz with a 40 mm hexagonal electrode at 730 V for 30 min. At 90-day follow-up, the lesion was completely replaced by eschar, with isolated tumor cells remaining in biopsies taken from the underlying vital tissue. No macroscopic disease was observed at 6 months, with significant improvement in quality of life (QoL). ECT for palliative purposes is also used for primary skin lesions with excellent aesthetic results and in terms of improving the quality of life [52,53].

Several studies have evaluated the impact of ECT on patients’ quality of life (QoL). Riva et al. observed, in a population with head and neck cancer, an objective tumor response rate of 48% (11% CR, 37% PR), with bleeding control achieved in 100% of patients and an improvement in quality of life. In particular, they noted improvements in general health status and social functioning, along with reductions in pain, the use of pain medication, and loss of appetite [54]. The prospective study by Rozsa et al. assessed 62 patients treated with ECT, with a median follow-up of 47 days. Before treatment, 38.7% of patients reported pain/discomfort and 24% reported anxiety/depression. After treatment, these percentages decreased to 32.2% and 19%, respectively, with a significant reduction in pain according to the VAS scale [53].

Across the included studies, electrochemotherapy was predominantly delivered with bleomycin (intravenous or intratumoral) according to contemporary operating procedures. Intralesional cisplatin-based ECT has also been employed and may be considered in selected cases, particularly when limiting systemic exposure is desirable; this approach is supported by primary clinical evidence in breast cancer cutaneous lesions, including a comparative, nonrandomized lesion-level study [20].

In the previously cited studies, electrochemotherapy (ECT) has emerged as a highly effective and well-tolerated local treatment, achieving high objective response rates, particularly in patients with smaller lesions (<3 cm), with lower toxicity compared to other strategies and a favorable impact on quality of life. Evidence from both prospective and retrospective studies reports overall response rates ranging between 70% and 90%, with complete response rates up to 58–60% in the most favorable subgroups. Efficacy appears to be influenced by several factors, including lesion size, receptor status, tumor grade, and concomitant systemic treatments. Receptor status has been reported to exert a differential impact on response rates across studies. Specifically, the INSPECT trial did not demonstrate statistically significant differences in overall response (OR) among the various histopathological subtypes. In contrast, analyses conducted by Cabula, as well as by Wichtowski and Russano, reported a significantly higher response rate in Luminal A and B carcinomas compared with triple-negative tumors. These apparent discrepancies may be attributed to the retrospective design and the relatively limited sample sizes of these studies. Prospective, adequately powered investigations are therefore warranted to resolve this controversy and to better delineate which patient subgroups are most likely to derive substantial benefit from this therapeutic approach. The differential response observed among these subtypes may be partly explained by the lower tumor aggressiveness typically associated with luminal tumors, which often present with smaller lesion sizes and lower Ki-67 indices, a potentially increased expression of the bleomycin-hydrolase enzyme in triple-negative subtypes and a higher degree of necrosis within triple-negative metastases. However, these findings warrant further investigation to better elucidate the underlying biological mechanisms driving the improved response rates.

Importantly, ECT has proven to be both safe and effective in elderly patients, a population often excluded from systemic therapies, further supporting its role in complex clinical and palliative settings. Despite methodological heterogeneity and the inherent limitations of retrospective analyses, the overall body of evidence supports ECT as a valuable therapeutic option for cutaneous breast cancer metastases, both in terms of local disease control and quality of life improvement. Moreover, the potential integration of ECT with systemic therapies or innovative approaches opens new avenues for clinical application.

For clarity and to facilitate direct comparison across studies, the main clinical trials investigating electrochemotherapy in cutaneous breast cancer metastases are summarized in Table 1, which provides a concise overview of patient numbers, study design, clinical response rates, and follow-up durations.

In conclusion electrochemotherapy (ECT) has progressively emerged as a promising and well-tolerated therapeutic option for the management of cutaneous and subcutaneous breast cancer metastases. Across retrospective and prospective studies, ECT demonstrates consistently high objective response rates, particularly in patients with smaller lesions (<3 cm) and in early-stage disease, while maintaining a favorable safety profile and improving patients’ quality of life. Its efficacy appears to be influenced by several clinical and biological factors, including lesion size, tumor receptor status, and the integration with systemic therapies. Although most evidence supports a higher response rate in Luminal A and B subtypes compared with triple-negative breast cancer, further prospective, adequately powered studies are warranted to clarify these differences and define the optimal patient selection criteria.

Importantly, ECT has shown notable effectiveness even in elderly and frail patients, a population frequently excluded from systemic therapeutic strategies, thus broadening its applicability in both curative and palliative contexts. The potential for combining ECT with systemic agents or innovative approaches highlights an expanding field of translational research that may further enhance its clinical utility. Despite the inherent limitations of the currently available studies—including methodological heterogeneity and the predominance of retrospective designs—the cumulative evidence supports ECT as a valuable, minimally invasive, and effective treatment modality for local disease control and quality-of-life improvement in patients with breast cancer cutaneous metastases.

## 4. Comparison with Other Therapeutic Options

The main locoregional techniques for treating cutaneous metastases are surgery, radiotherapy, photodynamic therapy, and electrochemotherapy. Immunotherapy and targeted agents are systemic treatments whose impact on cutaneous lesions is context-dependent and not directly comparable with procedures aimed at local control. Systemic chemotherapy remains standard for disseminated disease; systemic ORR in mBC varies by regimen and subtype, with cutaneous control depending on overall disease dynamics [55,56]. Radiotherapy (RT) is a valuable option, particularly for metastases located in areas that are difficult to operate on, providing meaningful pain relief and tumor size reduction. Response rates range from 60% to 80% for superficial lesions but are lower for deeper metastases [57]. Furthermore, multiple treatment sessions are usually required, and complete responses are relatively uncommon. Photodynamic therapy (PDT) offers a non-invasive option for superficial and visible metastases, with minimal toxicity such as pain or erythema at the treatment site. Nevertheless, available data are limited, with reported response rates between 50% and 70% but lacking randomized controlled trials (RCTs). Its use is restricted to superficial disease and is not applicable to all tumor types [58]. Surgery remains the most radical approach for well-defined, localized metastases, ensuring high response rates of up to 90%. However, the main drawbacks include the invasiveness of the procedure, risk of recurrence if excision is incomplete, and aesthetic or functional sequelae such as scarring or wound complications [59].

Electrochemotherapy (ECT), by combining electroporation with cytotoxic drugs, enhances intracellular drug uptake at lower doses and has shown remarkable efficacy in superficial and localized metastases. Reported clinical response rates reach 70–80%, with complete response (CR) achieved in approximately 57.5% of cases. Side effects are generally local and manageable, although efficacy decreases significantly in deep-seated lesions or those larger than 3 cm in diameter.

When therapeutic alternatives are limited, ECT could represent a good local treatment option for disease control. ECT can be used in areas that have already been pre-treated and it also can reach sensitive locations, such as the face. The temporary pore formation in the membrane with the ECT technique makes it more permeable to molecules that are otherwise poorly permeable or impermeable. Compared to chemotherapy, ECT requires lower drug doses administration (e.g., Bleomycin and Cisplatin) while achieving greater effectiveness in tumor tissue. Additionally, ECT reduces blood flow, allowing for a prolonged cytostatic effect within the tumor tissue [26].

In summary, while surgery and ECT represent the most effective strategies for localized cutaneous metastases, CT and RT are more suitable for systemic or complex disease scenarios, and PDT may serve as a complementary option in selected cases.

The comparison of the main therapeutic options for the treatment of cutaneous metastases, including their advantages, limitations, and response rates, is summarized in Table 2.

## 5. Challenges and Future Directions

Electrochemotherapy (ECT) has proven to be an effective and safe treatment for cutaneous metastases. The varying response rates reported in the literature can be attributed to factors such as the size of the treated lesions, the type of primary tumor, the procedures used, and the patient’s characteristics. For this reason, it is essential to standardize the treatment, carefully select patients, and determine the appropriate timing for its administration.

Regarding patient selection, ECT has been shown to be less effective in lesions larger than 3 cm in diameter, deeper lesions, or those in difficult-to-treat areas, according to several studies. Additionally, the best response rates have been observed in cutaneous metastases from melanoma and breast cancer. The patient’s immune status may also be a predictive factor for response. In fact, patients with compromised immune systems or elderly patients seem to have a poorer response to ECT, likely due to a reduced Abscopal effect. Other factors that appear to influence the response to ECT include a BMI lower than 22.5 kg/m^2^, a body surface area under 1.77 m^2^, and advanced stages of the primary tumor. Regarding breast cancer, reduced efficacy has been observed in ER+ patients, while higher efficacy has been noted in PgR− and HER2− patients.

Concerning the standardization of the procedure, ECT is typically administered according to a shared protocol (European Standard Operating Procedures of Electrochemotherapy, ESOPE), and the local response is evaluated according to the modified Response Evaluation Criteria in Solid Tumors (RECIST). However, there is no universally accepted protocol for ECT use, leading to variability in the approaches used, which complicates the comparison of results. Standardization should provide clear guidelines on the duration, voltage, and frequency of the applied pulses, as well as on the devices and electrodes to be used. Furthermore, precise instructions should be given on the type of chemotherapy to be used, the appropriate dosage, and the method of administration.

In the future, further research should focus on the integration of ECT with other therapies such as chemotherapy (CT), radiotherapy (RT), or immunotherapy, as the data available in the literature remain limited. Determining the best combination and the most effective sequencing of these treatments could lead to even higher response rates, positively impacting patients’ quality of life while minimizing side effects. Additionally, understanding how ECT interacts with these therapies at the molecular level could further optimize outcomes.

Future perspectives may also focus on the development of targeted electrode systems guided by imaging, which would enable more precise targeting to minimize damage to surrounding healthy tissue. The use of less invasive electrodes could also reduce adverse effects for patients. Research is also investigating new non-invasive or minimally invasive electroporation techniques, such as transdermal electroporation, which could be useful in the treatment of superficial tumors.

The combination of immunotherapy with ECT (e.g., pembrolizumab or nivolumab) could potentially enhance the immune response against tumors. Essentially, immune checkpoints are receptors expressed on the surface of T-cells and are responsible for negatively regulating the T-cell mediated immune response during the cancer immune-editing process. Immune checkpoint inhibitors (ICIs) suppress these receptors in order to reactivate the immune response against tumor cells. Preliminary evidence suggests that combining ECT with immune checkpoint inhibitors is feasible and may enhance local control; however, in breast cancer the data are limited to very small, non-comparative cohorts, precluding firm conclusions.

ECT can trigger local immunogenic cell death and antigen release. Whether the addition of ICIs translates into clinically meaningful improvements in breast cancer remains unclear, as available data are limited and non-comparative [46]. Personalized medicine approaches, in which ECT is combined with targeted therapies based on the specific genetics of the tumor, are also a promising direction. For example, tumors with specific mutations, such as BRAF mutations in melanoma, may respond better to certain therapeutic combinations. Preliminary, very small series suggest that combining ECT with immune checkpoint inhibitors is feasible and may enhance local control; in breast cancer, evidence is limited to non-comparative, very small cohorts, precluding firm conclusions. However, only a single published report has assessed this combination in breast cancer [51]. In that study, 55 patients with metastatic breast cancer underwent ECT at a median of 78 months following initial diagnosis. The complete response (CR) rate was 64%, with partial response (PR) in 22% and stable disease (SD) in 14%. Twenty-four-month progression-free survival (PFS) and overall survival (OS) were superior in the immunotherapy combination group, though this survival advantage was not maintained at 36 months. It is important to note that only three patients received the ECT + immunotherapy regimen in this cohort, highlighting the need for larger prospective studies to validate these findings. Preliminary, very small series suggest that combining ECT with ICIs is feasible and may enhance local control; however, data in breast cancer are limited to few patients and non-comparative designs, and no firm conclusions can be drawn. Prospective controlled studies are needed.

In the future, ECT could pave the way for more personalized therapeutic strategies. Advances in molecular diagnostics and imaging technologies may enable the identification of biomarkers that predict which patients will respond best to ECT. These biomarkers could assist in patient selection and optimize drug dosages, making treatment more effective while reducing side effects. Moreover, studying the immune system’s response to ECT could provide insights into how to enhance both local and systemic immune activation, thereby improving long-term outcomes.

## 6. Conclusions

In this scenario, early diagnosis is essential to treat patients promptly; for this reason specialists who deal with these pathologies should be well-informed about them [17]. Cutaneous metastases represent a significant challenge in the management of advanced-stage cancers. Among them, cutaneous metastases from breast cancer are the most frequent, alongside those from melanoma. Given the high incidence of breast cancer, they pose a particularly important issue for the national healthcare system. The prognosis for patients with metastatic breast cancer is generally unfavorable, with a 5-year survival rate of around 25%. However, the discovery of new therapeutic lines and an improved understanding of the biology of this cancer are gradually leading to a significant increase in survival rates. This, in turn, results in a higher likelihood of developing cutaneous metastases and a greater need to treat them effectively in order to ensure the best quality of life (QoL) for patients. Electrochemotherapy (ECT) has proven to be effective and safe in treating these metastases in addition to systemic treatment. Response rates in the literature vary based on factors such as lesion size, the stage of the primary tumor, and patient characteristics. However, the response is generally good, with reported overall response rates (ORR) of up to 70–80%.

The reported response rates across published studies remain highly heterogeneous, with complete response (CR) rates ranging from 8% to 77% in patients with cutaneous breast cancer metastases. Nonetheless, the majority of studies consistently report CR rates of 40–70% and overall response rates (ORRs) between 70% and 92%. Such variability may, at least in part, be attributed to differences in study design, as most available evidence derives from observational investigations, both retrospective and prospective. Sample size represents an additional confounding factor, with eight studies including fewer than 100 patients.

Among patient- and lesion-related variables, metastasis size emerged as the strongest determinant of treatment efficacy. The absence of randomization with respect to lesions larger or smaller than 3 cm may, therefore, have contributed to lower overall response rates in some analyses. Moreover, the histopathological subtype of the primary tumor has also been associated with differential treatment outcomes.

In all studies reviewed, electrochemotherapy (ECT) was performed according to the ESOPE protocol; however, treatment regimens varied, with some trials employing bleomycin alone and others adopting mixed approaches using either bleomycin or cisplatin. Notably, one study combined ECT with systemic therapy, introducing a potential source of bias in the interpretation of response outcomes.

Standardizing the treatment, including selecting the appropriate chemotherapy and personalizing the dose, is essential to optimize outcomes. The integration of ECT with other systemic therapies, such as chemotherapy, radiotherapy, or immunotherapy, could further improve response rates and patients’ quality of life.

The main limitations of this review relate to the scarcity of high-quality evidence in the current literature and the limited availability of randomized controlled trials.

Definitions of response and outcome measures are heterogeneous across studies, and long-term survival data are largely lacking. While ECT shows promising efficacy and a favorable safety profile, these results should be interpreted with caution. Consequently, ECT should currently be considered an emerging therapeutic modality rather than a definitive standard of care. Further prospective and randomized clinical trials are required to clarify optimal treatment regimens, identify predictors of response, and evaluate long-term outcomes, including survival and quality of life.

## Figures and Tables

**Figure 1 biomedicines-13-02345-f001:**
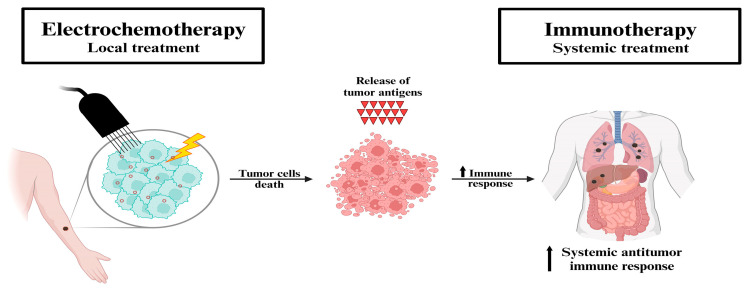
Schematic representation of the mechanism of electrochemotherapy. Proposed mechanism of the synergistic effect of electrochemotherapy and immunotherapy. Electrochemotherapy (local treatment) induces the immune system through immunogenic cell death (i.e., in situ vaccination). The tumor antigens released by the destroyed cells trigger a systemic antitumor immune response, which in turn is further exploited by immunotherapy (systemic treatment) in patients with metastatic disease. Source from Ref. [46]: Hadzialjevic, B.; Omerzel, M.; Trotovsek, B.; Cemazar, M.; Jesenko, T.; Sersa, G.; Djokic, M. Electrochemotherapy combined with immunotherapy—a promising potential in the treatment of cancer. *Front Immunol.*
**2024**, *14*, 1336866. https://doi.org/10.3389/fimmu.2023.1336866. Licensed under CC BY.3.1. Efficacy of Electrochemotherapy in Cutaneous Metastases of Breast Carcinoma.

**Table 1 biomedicines-13-02345-t001:** Summary of principal clinical studies on electrochemotherapy (ECT) for breast cancer cutaneous metastases.

Author, Year	Study Design	No. of Patients	Method/Protocol	Median Follow-Up	Risk of Bias	Response Criteria (RECIST or Other)	CR (%)	ORR (%)
Reberšek et al., 2004 [20]	Comparative, nonrandomized clinical study at the lesion level	6	Cisplatin	NR	High	NR	33%	100%
Benevento et al., 2012 [21]	Prospective observational	12	Bleomycin-based ECT, ESOPE protocol	210 days	Moderate	Clinical criteria	75.3%	92.3%
Matthiesen et al., 2012 [22]	Phase II, prospective	52	Bleomycin-based ECT, ESOPE protocol	12 months	Moderate	NR	8%	16%
Campana et al., 2012 [23]	Phase II, prospective	35	Bleomycin-based ECT, ESOPE protocol	36 months	Moderate	NR	54%	91%
Campana et al., 2014 [24]	Retrospective analysis	55	Bleomycin-based ECT	32 months	Moderate–High	NR	40%	87%
Cabula et al., 2015 [25]	Multicenter cohort	113	Bleomycin-based ECT	12 months	Moderate	Clinical criteria (RECIST-like)	58%	90%
Kreuter et al., 2015 [26]	Retrospective multicenter analysis	56	Bleomycin-based ECT	NR	Moderate–High	NR	10.7%	44.6%
Bourke et al., 2017 [27]	Retrospective single-center	24	Bleomycin-based ECT	10 years	Moderate	NR	64%	80%
Grischke et al., 2017 [28]	Observational study	33	Bleomycin-based ECT in addition to systemic therapy	3 years	Moderate	NR	58.4%	90.2%
Matthiensen et al., 2018 [22]	Prospective, multicenter observational registry study	90	Bleomycin-based ECT	NR	Moderate	NR	50%	71%
Wichtowski et al., 2019 [29]	Multicenter experience	38	Bleomycin-based ECT	NR	Moderate	NR	42%	71%
Clover et al., 2020 [30]	Pan-European registry (987 pts, various cancers, incl. BC)	300 (subgroup with BC)	Mixed (Bleomycin/Cisplatin)	NR	Moderate	NR	62–77%	85%
Russano et al., 2021 [31]	Retrospective Italian cohort	125	Bleomycin-based ECT, various protocols	5 years	Moderate–High	RECIST 1.1	40%	70–80%
Di Prata et al., 2023 (INSPECT) [32]	Prospective registry	171	Bleomycin, stratified by receptor status	12 ± 18 months	Moderate	Clinical response (INSPECT registry criteria)	58%	76–86%
Russano et al., 2025 [33]	Prospective multicentre study	195	Bleomycin, stratified by receptor status	12 months	Low–Moderate	Objective response (likely RECIST)	55%	82%

BC: breast cancer, CR: complete response, ORR: overall rate response, ECT: electrochemotherapy.

**Table 2 biomedicines-13-02345-t002:** Locoregional and systemic options for cutaneous metastases from breast cancer.

Therapeutic Option	Mechanism	Advantage	Limitation	Main Adverse Effect (Safety)	Outcomes (BC Cutaneous Metastases)
**Systemic Therapy** (CT/ET/Targeted/ICI) [1,2,55,56]	Cytotoxics, endocrine agents, targeted drugs, and immunotherapy act systemically on tumour burden.	Treats disseminated disease; may shrink cutaneous lesions as part of overall response.	Cutaneous control is context-dependent; discordance with systemic response can occur; drug-specific toxicities.	Drug class-specific adverse effects (e.g., myelosuppression, mucositis, hand–foot syndrome, immune-related AEs).	NR (breast cutaneous lesion-level CR/ORR not consistently reported in systemic trials; responses vary by subtype and regimen).
**ECT** [7,8,12,22,24,25,26,29,31,32,33,39,51]	Transient permeabilization by electric pulses to enhance uptake of cytotoxic agents (bleomycin IV/IT; cisplatin IT).	High local control in superficial lesions; outpatient/sedation feasible; repeatable; spares systemic exposure (IT).	Limited depth (few cm) and coverage for bulky /infiltrative disease; requires electrode access and pain control; outcomes decline with large (>3 cm) lesions.	Pain during pulses; transient erythema/edema; ulceration/wound complications in previously irradiated fields; typically low systemic toxicity.	Patient-level (12 weeks): ORR ~71%, CR ~42% (Wichtowski 2019 [29]). Patient-level (~2 months): ORR 71%, CR 50% (Matthiessen 2018, INSPECT [51]). Lesion-level: CR 75.3%, ORR 92.3% (Benevento 2012) [21].
**RT** [6,57]	Ionizing radiation for local control/palliation.	Non-invasive; broad field coverage; haemostasis and pain relief.	Constraints with prior high-dose irradiation; radioresistance in heavily pre-treated fields; dermatitis and wound-healing issues.	Dermatitis, fatigue; risk of ulceration/fibrosis in previously irradiated skin	ORR: 60–80%, CR: NR (local control varies with dose, fractionation, and prior RT).
**Photodynamic Therapy** (PDT) [58]	Photosensitizer + light activation → reactive oxygen species and tumour necrosis.	Selective destruction of superficial lesions; tissue-sparing; repeatable.	Limited depth (mm); pain during illumination; photosensitivity precautions; limited evidence in breast cutaneous metastases.	Pain during illumination; photosensitivity; local erythema/necrosis.	50–70% (few breast-specific series; outcomes depend on lesion thickness and illumination parameters; no RCTs available).
**Surgery** [59]	Excision of isolated/symptomatic lesions; reconstruction as needed.	Definitive removal for limited disease; immediate symptom relief; pathology confirmation.	Not suitable for multifocal/field disease; morbidity of wide excisions; recurrence risk on irradiated chest wall.	Surgical site complications, dehiscence/infection; anaesthesia-related risks.	Up to 90% (when feasible in selected localized lesions; outcomes depend on margins and disease extent).

## Data Availability

Not applicable.

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
