# Peer review of "The Role of Electrochemotherapy in Managing Cutaneous Metastases from Breast Carcinoma: A Narrative Review"

_biomedicines, 2025, doi:10.3390/biomedicines13102345_

Round 1

Reviewer 1 Report

Comments and Suggestions for Authors

The manuscript submitted by Russano et al. provides an extensive review of the use of electrochemotherapy for the treatment of cutaneous metastases derived from breast carcinoma. The authors present a compilation of clinical evidence, including retrospective and prospective studies, as well as multicenter experiences. However, I consider that the review falls short in terms of critical analysis, physiopathological integration, and the proposal of mechanistic hypotheses. The review lacks a comparative discussion between studies with divergent findings, does not explore the tumor biology in depth that could explain the observed responses to ECT, and omits the inclusion of several recent and relevant studies. The writing shows repetitions, some grammatical errors, and lack of cohesion in certain sections. Additionally, I believe that some figures and tables add minimal analytical value.

Major Comments

-Most of the text is purely descriptive and reproduces results from multiple studies without any critical discussion of their strengths or limitations. The manuscript does not address possible physiopathological explanations as to why certain molecular subtypes of breast cancer respond better to ECT (example., luminal A/B vs. triple (-)).

-Nor does it discuss the role of the tumor microenvironment or tumor immunogenicity in the response to ECT. It is recommended to integrate some hypotheses based on the immunomodulatory mechanisms of ECT, such as the abscopal effect or the activation of specific T cells, especially in combination with immunotherapy.

-The authors cite multiple studies with very variable response rates (CR from 8% to 85%) but do not critically analyze these differences. A discussion on methodological factors that could explain this variability is missing (like inclusion criteria, tumor size, anatomical location, bleomycin dose, electric pulse parameters, operator experience, etc.). It is imperative to introduce a comparative methodological synthesis table to facilitate interpretation across studies.

-Although the ms cites a significant number of studies, most of them are retrospective in design. There is a lack of mention of controlled clinical trials or recent prospective studies (preferably after 2022), including those investigating combinations with immunotherapy. For instance, studies on ECT combined with immune checkpoint inhibitors like anti-PD-1 (nivolumab) should at least be mentioned as emerging lines of research.

-The only figure presented (Fig1) shows clinical variability but without detailed legends, scales, or clinical context. In my opinion, this image does not contribute substantially to critical analysis. Table 1 contains typographical errors (“ECT” described as “electrochemical therapy” instead of electrochemotherapy), confusing juxtaposition of data, and lacks references to support the “Response Rates” values. I suggest redesigning this table by incorporating numerical references, methodology of data extraction, and standardized definitions (ORR according to RECIST v1.1).

-The quality of writing is inconsistent. There are fragments with grammatical mistakes (“therapy with electrochemical therapy”) and redundant constructions (“significant improvement in quality of life… improving the quality of life”). Also, informal phrases are observed (“it constitutes a sticking point for surgeons”), which are unclear in the context of a scientific review and should be replaced with precise technical language. Paragraph structure also presents cohesion problems: results, methodology, and conclusions are often mixed in the same block, making critical reading and for this reviewer, interpretation difficult.

-Although the possibility of combining ECT with curcuminoids is briefly mentioned, the proposal lacks robust experimental support and is based on a single in vitro study. The authors does not discuss the biopharmaceutical challenges of combining ECT with compounds like curcumin (bioavailability, stability, distribution…). It is advisable to discuss more relevant and validated combinations in preclinical or clinical models, such as ECT + checkpoint inhibitors or ECT + antiangiogenic agents. Studies in immunocompetent murine models could also be proposed to explore activation of specific immune responses.

-Each section ends abruptly without a summary or partial conclusion integrating key points and their link to subsequent sections. This structure limits overall understanding for the reader. It is recommended to close each section with a short synthesis paragraph highlighting the main findings, limitations of the evidence, and potential clinical implications..

-I consider it necessary to add a schematic figure (or figures) showing the mechanism of action of ECT, integrating the steps of electroporation, bleomycin uptake, and immune activation.

-A section about the limitations of the reviewed literature should be included, emphasizing the lack of high-level evidence studies consistent with the scope of a comprehensive review.

Minor Comments

The English writing should be completely revised to improve cohesion and clarity.

The references must be reformatted to avoid duplicate numbering or missing entries (some are out of order).

Comments on the Quality of English Language

The quality of writing is inconsistent. There are fragments with grammatical mistakes (“therapy with electrochemical therapy”) and redundant constructions (“significant improvement in quality of life… improving the quality of life”). Also, informal phrases are observed (“it constitutes a sticking point for surgeons”), which are unclear in the context of a scientific review and should be replaced with precise technical language. Paragraph structure also presents cohesion problems: results, methodology, and conclusions are often mixed in the same block, making critical reading and for this reviewer, interpretation difficult.

Author Response

Reviewer #1 (Comments to the Author):

The manuscript submitted by Russano et al. provides an extensive review of the use of electrochemotherapy for the treatment of cutaneous metastases derived from breast carcinoma. The authors present a compilation of clinical evidence, including retrospective and prospective studies, as well as multicenter experiences. However, I consider that the review falls short in terms of critical analysis, physiopathological integration, and the proposal of mechanistic hypotheses. The review lacks a comparative discussion between studies with divergent findings, does not explore the tumor biology in depth that could explain the observed responses to ECT, and omits the inclusion of several recent and relevant studies. The writing shows repetitions, some grammatical errors, and lack of cohesion in certain sections. Additionally, I believe that some figures and tables add minimal analytical value.

Response:

We thank the reviewer for the feedback and the opportunity to further complete by integrating and expanding the indicated sections.

Specifically:

Major Comments

-Most of the text is purely descriptive and reproduces results from multiple studies without any critical discussion of their strengths or limitations. The manuscript does not address possible physiopathological explanations as to why certain molecular subtypes of breast cancer respond better to ECT (example., luminal A/B vs. triple (-)). 

RE: Done as request lines  (372-387)

-Nor does it discuss the role of the tumor microenvironment or tumor immunogenicity in the response to ECT. It is recommended to integrate some hypotheses based on the immunomodulatory mechanisms of ECT, such as the abscopal effect or the activation of specific T cells, especially in combination with immunotherapy.

RE: Done as request lines  (163-173)

-The authors cite multiple studies with very variable response rates (CR from 8% to 85%) but do not critically analyze these differences. A discussion on methodological factors that could explain this variability is missing (like inclusion criteria, tumor size, anatomical location, bleomycin dose, electric pulse parameters, operator experience, etc.). It is imperative to introduce a comparative methodological synthesis table to facilitate interpretation across studies.

RE: Done as request lines  and We have revised and updated the Table.1 following the instructions.

-Although the ms cites a significant number of studies, most of them are retrospective in design. There is a lack of mention of controlled clinical trials or recent prospective studies (preferably after 2022), including those investigating combinations with immunotherapy. For instance, studies on ECT combined with immune checkpoint inhibitors like anti-PD-1 (nivolumab) should at least be mentioned as emerging lines of research.

RE: Done as request we have revised the section 4 lines (423-461) e and created the Table.2

-The only figure presented (Fig1) shows clinical variability but without detailed legends, scales, or clinical context. In my opinion, this image does not contribute substantially to critical analysis. Table 1 contains typographical errors (“ECT” described as “electrochemical therapy” instead of electrochemotherapy), confusing juxtaposition of data, and lacks references to support the “Response Rates” values. I suggest redesigning this table by incorporating numerical references, methodology of data extraction, and standardized definitions (ORR according to RECIST v1.1).

RE: Figure 1 has been removed from the manuscript, and Table 1 has been completely revised, corrected, and supplemented.

-The quality of writing is inconsistent. There are fragments with grammatical mistakes (“therapy with electrochemical therapy”) and redundant constructions (“significant improvement in quality of life… improving the quality of life”). Also, informal phrases are observed (“it constitutes a sticking point for surgeons”), which are unclear in the context of a scientific review and should be replaced with precise technical language. Paragraph structure also presents cohesion problems: results, methodology, and conclusions are often mixed in the same block, making critical reading and for this reviewer, interpretation difficult.

RE:  We have corrected the manuscript throughout, trying to use more appropriate technical and scientific language.

-Although the possibility of combining ECT with curcuminoids is briefly mentioned, the proposal lacks robust experimental support and is based on a single in vitro study. The authors does not discuss the biopharmaceutical challenges of combining ECT with compounds like curcumin (bioavailability, stability, distribution…). It is advisable to discuss more relevant and validated combinations in preclinical or clinical models, such as ECT + checkpoint inhibitors or ECT + antiangiogenic agents. Studies in immunocompetent murine models could also be proposed to explore activation of specific immune responses.

RE: The reference to curcumin and the use of curcuminoids in combination with ECT has been excluded from the manuscript.

-Each section ends abruptly without a summary or partial conclusion integrating key points and their link to subsequent sections. This structure limits overall understanding for the reader. It is recommended to close each section with a short synthesis paragraph highlighting the main findings, limitations of the evidence, and potential clinical implications..

RE: We have tried to implement your suggestions by improving the flow of each paragraph.

-I consider it necessary to add a schematic figure (or figures) showing the mechanism of action of ECT, integrating the steps of electroporation, bleomycin uptake, and immune activation.

RE: Figure 1 is included in the text. This figure is taken from a publication that represents a milestone in ECT.

-A section about the limitations of the reviewed literature should be included, emphasizing the lack of high-level evidence studies consistent with the scope of a comprehensive review.

  1. In each paragraph and section, I have highlighted the limitations of the procedure, supporting the purpose of our review.

Minor Comments

The English writing should be completely revised to improve cohesion and clarity.

RE: I have tried to improve the English. If it is still not adequate, we will use the publisher's linguistic revision service.

The references must be reformatted to avoid duplicate numbering or missing entries (some are out of order).

RE: Done as request

Comments on the Quality of English Language

The quality of writing is inconsistent. There are fragments with grammatical mistakes (“therapy with electrochemical therapy”) and redundant constructions (“significant improvement in quality of life… improving the quality of life”). Also, informal phrases are observed (“it constitutes a sticking point for surgeons”), which are unclear in the context of a scientific review and should be replaced with precise technical language. Paragraph structure also presents cohesion problems: results, methodology, and conclusions are often mixed in the same block, making critical reading and for this reviewer, interpretation difficult

RE: I have tried to improve the English. If it is still not adequate, we will use the publisher's linguistic revision service.

Reviewer 2 Report

Comments and Suggestions for Authors

This article presents a synthetic and up-to-date analysis of the literature on electrochemotherapy (ECT) in the treatment of cutaneous metastases from breast cancer. It provides a valuable overview of clinical and experimental knowledge. It provides a broad clinical context, an overview of ECT's mechanisms of action, and an analysis of its effectiveness in various patient populations, taking into account predictive factors and future directions for the development of this technology. After reviewing the article, I have several comments:

- The article lacks a detailed description of the literature search methodology: it does not specify which databases were searched, what inclusion/exclusion criteria were used, or how the sources were selected.

- The absence of a PRISMA diagram, which is currently a common standard for systematic reviews.

- The absence of a quality assessment of the studies analyzed in the literature.

- I suggest adding a summary table comparing clinical trials (CR, ORR, n, method, follow-up time), which would facilitate comparisons.

Author Response

Reviewer #2 (Comments to the Author):

This article presents a synthetic and up-to-date analysis of the literature on electrochemotherapy (ECT) in the treatment of cutaneous metastases from breast cancer. It provides a valuable overview of clinical and experimental knowledge. It provides a broad clinical context, an overview of ECT's mechanisms of action, and an analysis of its effectiveness in various patient populations, taking into account predictive factors and future directions for the development of this technology. After reviewing the article, I have several comments:

Response:

We thank the reviewer for the feedback and the opportunity to further complete by integrating and expanding the indicated sections.

Specifically:

- The article lacks a detailed description of the literature search methodology: it does not specify which databases were searched, what inclusion/exclusion criteria were used, or how the sources were selected.

- The absence of a PRISMA diagram, which is currently a common standard for systematic reviews.

- The absence of a quality assessment of the studies analyzed in the literature.

- I suggest adding a summary table comparing clinical trials (CR, ORR, n, method, follow-up time), which would facilitate comparisons.

RE: We explained the literature search methodology, created the Prisma Flow Figure, and revised Table 1 according to your instructions.

Reviewer 3 Report

Comments and Suggestions for Authors

The review introduces the principles of electrochemotherapy (ECT), its mechanisms of action, and clinical efficacy in cutaneous metastases of primary breast cancer patients. Authors did a good job of compiling the recent publications in the field, more than 30% of the citations are from the last five years. Although there are reviews in ECT, the focus on the clinical efficacy in cutaneous metastases of breast cancer gives a distinct niche to the review.

A few specific comments:

In the section Efficacy of Electrochemotherapy in Cutaneous Metastases of Breast Carcinoma, the authors present comprehensive review of the current ECt status and their clinical efficacy. However, the conclusion of all the listed studies is missing. Some of it is discussed in the challenges and future directions section, which can also be touched in this section.

In the section comparison with other therapeutic options, the authors list few therapeutic options used to treat melanoma/breast cancer cutaneous metastases but don’t really compare the efficacy. Most of the comparison is shown in the table but not discussed in the text. Table 1 stands on its own without any mention in the text. Neoantigen vaccines are coming up as a new therapeutic modality for in transit metastases and unresectable lesions, which can also be discussed.

Lines 45-49 Different fonts used

Line 58-59 This is the last sentence of the intro, and it provides mortality rate of stage IV breast cancer, which is a bit confusing as the review is about the skin metastases. The last few sentences of the intro should highlight why skin metastases of the breast cancer should be reviewed and why electrochemotherapy is worth considering.

Line 80 Citation is in between two full stops

Line 118 the word paragonable is not the best choice of the word for this scenario

Line 302-327 Different font used

Line 330 Citation is in between two full stops

Author Response

Reviewer #3 (Comments to the Author):

The review introduces the principles of electrochemotherapy (ECT), its mechanisms of action, and clinical efficacy in cutaneous metastases of primary breast cancer patients. Authors did a good job of compiling the recent publications in the field, more than 30% of the citations are from the last five years. Although there are reviews in ECT, the focus on the clinical efficacy in cutaneous metastases of breast cancer gives a distinct niche to the review.

Response:

We thank the reviewer for the feedback and the opportunity to further complete by integrating and expanding the indicated sections.

Specifically:

In the section Efficacy of Electrochemotherapy in Cutaneous Metastases of Breast Carcinoma, the authors present comprehensive review of the current ECt status and their clinical efficacy. However, the conclusion of all the listed studies is missing. Some of it is discussed in the challenges and future directions section, which can also be touched in this section.

RE: done as requested lines (398-417)

In the section comparison with other therapeutic options, the authors list few therapeutic options used to treat melanoma/breast cancer cutaneous metastases but don’t really compare the efficacy. Most of the comparison is shown in the table but not discussed in the text. Table 1 stands on its own without any mention in the text. Neoantigen vaccines are coming up as a new therapeutic modality for in transit metastases and unresectable lesions, which can also be discussed.

RE: we completely rewrote the section and designed Table 2, confirming your requests.

Lines 45-49 Different fonts used

RE: corrected

Line 58-59 This is the last sentence of the intro, and it provides mortality rate of stage IV breast cancer, which is a bit confusing as the review is about the skin metastases. The last few sentences of the intro should highlight why skin metastases of the breast cancer should be reviewed and why electrochemotherapy is worth considering.

RE: We  completely revised the sentence and integrated lines (59-69)

Line 80 Citation is in between two full stops

RE: WE have corrected  

Line 118 the word paragonable is not the best choice of the word for this scenario

RE: WE have corrected  

Line 302-327 Different font used

RE: WE have corrected  

Line 330 Citation is in between two full stops

RE: WE have corrected  

Round 2

Reviewer 1 Report

Comments and Suggestions for Authors

The authors have made changes to their review, but, the main corcern in the manuscript is that is presented as a narrative review (or just called review), but at the same time the authors state that they followed PRISMA 2020 guidance, applied eligibility criteria, study quality assessment tools, and even constructed a PRISMA flow diagram. This creates an important inconsistency. If PRISMA methodology and critical appraisal of studies are applied, the work cannot be considered a purely narrative review. Rather, it should be classified as a systematic review with qualitative synthesis (without meta-analysis). Presenting it as “narrative” while using PRISMA elements undermines methodological clarity and may mislead readers about the rigor of the evidence synthesis. The authors should clarify the design and align the terminology: either explicitly call it a systematic review (qualitative) or, if they prefer to maintain the narrative label, they should remove PRISMA terminology and flow diagram. At minimum, the rationale for mixing approaches should be transparently justified.

Other major comments
As previosly pointed in first review  round, the review includes a large number of retrospective and prospective cohort studies on ECT, but the synthesis is highly descriptive and repetitive. Several paragraphs reproduce response rates from small cohorts without critical interpretation. A more analytical comparison across studies, stratified by tumor size, receptor subtype, and prior radiotherapy, would provide added value. In my opinión, the current narrative feels more like a catalog of studies than a critical synthesis.

Although the authors mention that quality assessment tools (ROBINS-I, Newcastle–Ottawa, Cochrane RoB) were applied, there is no table or clear summary of risk-of-bias results. This is a missed opportunity. Presenting a structured assessment of study quality would strengthen the credibility of the conclusions. Right now, readers could cannot discern whether the positive results derive mainly from low-bias prospective cohorts or from small, retrospective, high-bias series.

Table 1 and 2 are missing in the main text of ms and the second table is missing of references..

The PRISMA diagram is presented, but given the methodological inconsistency mentioned above, its inclusion should be reconsidered.

Table 1, summarizing clinical trials, is useful but could be expanded to include risk-of-bias judgments, median follow-up, and specific response evaluation criteria (RECIST vs other definitions).

The section on ECT mechanisms of action (immune effects, abscopal effect) is comprehensive, but it is poorly integrated with the clinical synthesis. For example, if immunogenic cell death and abscopal responses are hypothesized, the review should discuss why no prospective trial has confirmed durable systemic effects in breast cancer. The disconnect between mechanistic rationale and clinical outcomes should be addressed explicitly.

The conclusion presents ECT almost as a standard of care, highlighting high response rates and safety. However, the authors do not sufficiently emphasize the lack of randomized controlled trials, the heterogeneity of definitions, and the absence of long-term survival data. Recommendations for clinical guidelines should be tempered, framing ECT as an emerging modality rather than a definitive standard.

minor

Language requires minor editing: some sections repeat “ECT has shown to be effective…” without variation, creating redundancy

Comments on the Quality of English Language

Language requires minor editing: some sections repeat “ECT has shown to be effective…” without variation, creating redundancy

Author Response

Reviewer #1 (Comments to the Author):

1.The authors have made changes to their review, but, the main corcern in the manuscript is that is presented as a narrative review (or just called review), but at the same time the authors state that they followed PRISMA 2020 guidance, applied eligibility criteria, study quality assessment tools, and even constructed a PRISMA flow diagram. This creates an important inconsistency. If PRISMA methodology and critical appraisal of studies are applied, the work cannot be considered a purely narrative review. Rather, it should be classified as a systematic review with qualitative synthesis (without meta-analysis). Presenting it as “narrative” while using PRISMA elements undermines methodological clarity and may mislead readers about the rigor of the evidence synthesis. The authors should clarify the design and align the terminology: either explicitly call it a systematic review (qualitative) or, if they prefer to maintain the narrative label, they should remove PRISMA terminology and flow diagram. At minimum, the rationale for mixing approaches should be transparently justified.

Response:

We thank the reviewer for this important comment. We agree with the observation: our manuscript is, to all effects, a narrative review (lines 3). Therefore, all references to systematic review methodology, including PRISMA guidance, the flow diagram, and study quality assessment, have been removed from the text to avoid any inconsistency and ensure methodological clarity

Other major comments
2. As previosly pointed in first review round, the review includes a large number of retrospective and prospective cohort studies on ECT, but the synthesis is highly descriptive and repetitive. Several paragraphs reproduce response rates from small cohorts without critical interpretation. A more analytical comparison across studies, stratified by tumor size, receptor subtype, and prior radiotherapy, would provide added value. In my opinión, the current narrative feels more like a catalog of studies than a critical synthesis.

Response:

We thank the reviewer for this insightful comment. We acknowledge that the synthesis is largely descriptive and that the available evidence mainly consists of small retrospective and prospective cohort studies, with considerable heterogeneity in patient populations, tumor characteristics, and prior treatments. Given these limitations, we deliberately chose a narrative approach, which allows us to summarize the state of the art without overinterpreting results that are not directly comparable. We agree that a more analytical synthesis stratified by tumor size, receptor subtype, and prior radiotherapy would be of great value; however, such an effort would require more homogeneous data or randomized controlled trials, which are not yet available. Once higher-level evidence emerges, we fully agree that a systematic and more structured comparison will be warranted.

  1. 3. Although the authors mention that quality assessment tools (ROBINS-I, Newcastle–Ottawa, Cochrane RoB) were applied, there is no table or clear summary of risk-of-bias results. This is a missed opportunity. Presenting a structured assessment of study quality would strengthen the credibility of the conclusions. Right now, readers could cannot discern whether the positive results derive mainly from low-bias prospective cohorts or from small, retrospective, high-bias series.

Response:

We thank the reviewer for this valuable observation and apologize for the inconsistency. We have removed all references to formal quality assessment tools (ROBINS-I, Newcastle–Ottawa, Cochrane RoB) to align the manuscript with its nature as a narrative review. We have completely removed paragraph 2.5 Quality assessment.In line with the reviewer’s suggestion, we have revised the table1 to include notes on the potential risk of bias of the included studies, in order to provide readers with a clearer understanding of the strength of the available evidence. We have revised the manuscript accordingly.

  1. Table 1 and 2 are missing in the main text of ms and the second table is missing of references.

Response:

Tables 1 and 2 have been reintegrated into the text: Table 1 has been completely revised and references have been added to Table 2.

  1. The PRISMA diagram is presented, but given the methodological inconsistency mentioned above, its inclusion should be reconsidered

Response:

The Prisma Diagram It has been completely removed.

  1. Table 1, summarizing clinical trials, is useful but could be expanded to include risk-of-bias judgments, median follow-up, and specific response evaluation criteria (RECIST vs other definitions).

Response:

 Table 1 has been completely revised, complying with all your instructions and requests.

  1. The section on ECT mechanisms of action (immune effects, abscopal effect) is comprehensive, but it is poorly integrated with the clinical synthesis. For example, if immunogenic cell death and abscopal responses are hypothesized, the review should discuss why no prospective trial has confirmed durable systemic effects in breast cancer. The disconnect between mechanistic rationale and clinical outcomes should be addressed explicitly.

Response:

We thank the reviewer for this suggestion. We have revised the section on ECT mechanisms of action to explicitly acknowledge the current gap between the strong immunological rationale (immunogenic cell death, abscopal effect) and the lack of prospective clinical trials confirming durable systemic responses in breast cancer. The text now emphasizes that, while preclinical data are promising, clinical evidence remains limited, and this disconnect should be taken into account when interpreting the available findings. We have revised the manuscript accordingly.

  1. The conclusion presents ECT almost as a standard of care, highlighting high response rates and safety. However, the authors do not sufficiently emphasize the lack of randomized controlled trials, the heterogeneity of definitions, and the absence of long-term survival data. Recommendations for clinical guidelines should be tempered, framing ECT as an emerging modality rather than a definitive standard.

Response:

We thank the reviewer for this  comment. We have revised the Conclusions section to provide a more balanced interpretation of the available evidence. While ECT demonstrates promising efficacy and a favorable safety profile, we now explicitly acknowledge the limitations of the current literature, including the scarcity of randomized controlled trials, the heterogeneity of outcome definitions, and the lack of long-term survival data. In light of these limitations, we have reframed ECT as an emerging therapeutic modality rather than a definitive standard of care. We believe this revision more accurately reflects the current state of evidence while highlighting areas where further research is needed. The manuscript has been revised accordingly.

Minor

  1. Language requires minor editing: some sections repeat “ECT has shown to be effective…” without variation, creating redundancy

Response:

 We have reviewed the quality of the English, but we are willing to use the journal's language editing service if the quality is still not adequate. 

Round 3

Reviewer 1 Report

Comments and Suggestions for Authors

The authors have made signofocant changes to their review and improve it